# Spatio-Temporal Fish Catch Assessments Using Fishing Vessel Trajectories and Coastal Fish Landing Data from around Jeju Island

**Solomon Amoah Owiredu** 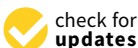 **and Kwang-Il Kim \***

College of Ocean Sciences, Jeju National University, Jeju 63243, Korea; saowiredu@jejunu.ac.kr
* Correspondence: kki@jejunu.ac.kr; Tel.: +82-64-754-3415

**Abstract:** While pressure on marine ecosystems leading to declines in global fish catches have been attributed to excessive fishing and to unregulated and unreported fishing, existing management practices have yet to fully address these declines. Estimation of spatial and temporal distribution of fisheries resources and the extent of fishing impacts on marine ecosystems using vessel trajectories has become central in recent studies. This study proposed the use of trajectories of 771 Korean coastal and offshore fishing vessels and one-year fish landing data to estimate variations in commercial fish species, vessel, and fishing gear activity distributions in the waters around Jeju island. A set of standards were applied to identify individual fishing tracks of major gears and uniformly distributed catch to fishing segments of trajectories to produce spatio-temporal distributions of catch, fishing activities, and vessel reliance on fishing grounds at a fine spatial scale. The method identified reference points that can inform management at local and regional scales. We discuss the opportunities of combining larger datasets collected over a longer period and applying predictive modeling techniques in making extensive assessments, including climate change impacts on fishing activities that can inform resource management and marine spatial planning.

**Keywords:** commercial fish stocks; fish population declines; illegal, unreported and unregulated (IUU) fishing; automatic identification system (AIS); overfishing

## 1. Introduction

Ocean ecosystems have continued to support our ever-increasing global demand for fish resources, and have provided critical services that support economies and improve livelihoods [1]. Over the years, the use of the marine environment has expanded at a rate that may pose a threat to the sustenance and survival of resources, and has generated conflicts among users. Marine stocks are decreasing due to overfishing, pollution, and the effects of global climate change. However, protecting the integrity of the maritime spaces to support multiple uses require a deeper understanding of ocean dynamics, the impacts of human activities, and the need to develop appropriate management structures that evolve with the changing environment. An improved understanding of the real spatial and temporal extents of fishing activities can help to enhance the effectiveness of management decisions implemented for the sustainable exploitation of resources [2].

Fishery production from Korean waters has been decreasing over the past two decades, due to overexploitation caused by overcapacity, climate change, destruction of marine ecosystems, illegal fishing by foreign vessels, and implementation of production-oriented fishery policies [3]. Over the previous 50 years (1968–2017), the surface seawater temperature of the Korean Peninsula has risen by approximately 1.23 °C (2.5 times faster than the global average of 0.48 °C), and this has intensified the volatility of fishery resources, and has led to concerns regarding fishery disasters, as they may occur with high water temperatures. Ref. [4] investigated the changes in marine ecosystems in the waters of Jeju Island caused by variations in environmental factors. The study identified long-term

changes in fish species composition and community structures due to temperature, salinity, and volume transport of the Tsushima Warm Current and the Korean Strait Bottom Cold Water by analyzing fish and environmental data obtained for Jeju waters from 1981 to 2010. Several resource management plans and programs have been implemented by the Korean government to address the decline in the fish landings, including measures such as artificial reefs and sea ranches, vessel reduction, total allowable catches (TAC), and fishery stock rebuilding projects. Despite these efforts, overfishing and illegal, unreported, and unregulated (IUU) fishing has undermined several of government's resource management programs aimed at the recovery of marine fish stocks [5,6].

To efficiently manage fishery resources and establish appropriate policies to ensure sustainability, it is important to quantitatively evaluate the spatial and temporal distribution of resources. However, most countries depend on the records of fish catch sales, direct collections onboard, and fishermen questionnaires for the assessment of their fishery resources. These assessment methods can only obtain single-type data for analysis, such as fish species and fishing gear type. Recently, automatic identification systems (AIS) were introduced into the fishing industry, and have aided in overcoming the limitations of analyzing existing fishing resources. AIS data provide fishing vessel trip data with high spatio-temporal resolution using the VHF radio frequency band [7]. The data facilitates the identification of fishing and non-fishing activities for a given location, but does not include the fish catch information with respect to quantity and location of harvest.

Catch information including weight, species, and fish condition at landing necessary for management purposes is primarily recorded at port, and is used by managers and regulators. In this study, a combination of AIS and fish landing data spanning the period from January to December 2018 was processed to assess the impacts of fishing activities on fishing grounds in the waters around Jeju island. Our work focused on coastal and inland fishing vessels that operated within the study area. Vessel owners often do not report their catch or ground information to the government as required. In addition, fishermen are hesitant in sharing fishing grounds information with other vessel owners. As fishing vessels are mandated to install a position indicating device, this data has become increasingly useful in estimating fishing activities and catch on the basis of vessel trajectory which is enriched with information including maritime mobile service identity (MMSI) (vessel identifier), departure time of the trip (exit from port area), departure port, position of the segment with respect to the port areas, average speed, and activity of the vessel within the segment of the fishing gear which can be obtained through the MMSI [8]. We obtained and processed raw AIS data and reconstructed trajectories of coastal and inland fishing vessels. Vessel positions were distinguished for fishing and non-fishing activities using speed profiles. In addition, we distributed fish catch among trajectories that were associated with fishing activities, and employed the uniform distribution technique where each vessel's catch is uniformly distributed along the segments of trajectory considered as fishing. We mapped these at fine spatial scales to assess the spatio-temporal extent, impact, and variations in the activities of these vessels.

Fisheries investigations have previously used a variety of different tracking devices to monitor the spatio-temporal dynamics of fishing vessels and to quantify fishing efforts and intensities in the fishing grounds and fisheries resources therein. Prior to the use of the AIS data, satellite-based vessel monitoring system (VMS) were used to monitor vessel activity and map fishing efforts at high spatial resolutions [9,10]. The introduction of VMS by the European Union (EU) resulted in significant advancements in vessel monitoring and the mapping of fishing efforts [11]. Ref. [12] provides an account of the characteristics of VMS data and their applications in fishery research and outline the opportunities that AIS data present for mapping the impacts of fishing activities on a global scale.

Although the spatial and temporal monitoring of vessel activity and the analyses of fishing effort intensities on marine ecosystems have been enhanced by VMS data globally [13], it is difficult to access the data due to confidentiality and concerns related to data protection [14]. The difficulties regarding VMS data in terms of identifying fishing activity

and gear type are due to the long intervals of vessel trajectory data [15], leading to AIS data, which is characterized by high persistence, being a better choice [8]. The availability of satellite AIS (S-AIS) data from a growing number of satellite-based data providers [16] has made it readily available for fisheries resource assessments.

AIS was originally intended to help improve ship safety and to transmit at high frequencies to avoid ship collisions. However, its expansion in recent years has increased its popularity in academic research and has now been practically applied in various disciplines. This is because it aids in addressing resource management challenges and provides data on a scale that is critical to ensure the sustainability of marine fishery resources. Although the spatial and temporal coverage of VMS and AIS data differ in availability, it has been recommended that the integrated use of both would present a more holistic and effective approach in the assessment of fishing pressures on marine ecosystems, by correcting the bias associated with the use of one data source [11].

Fishing types have been evaluated using vessel speed profiles. However, vessel operational speeds vary for different fishing gear types and vessel activities (fishing and non-fishing). Complex algorithms have been developed to distinguish among the different gear types and vessel activities. Ref. [17] developed algorithms to identify the fishing activities of trawlers (Hidden Markov Model), longliners (Data Mining), and purse seiners (multi-layered filtering strategy based on vessel speed and operation time) from satellite AIS data using data mining and machine learning. The variation in speed and distribution patterns for the vessel types (gear type) determines which model (algorithm) to use in determining and mapping fishing activities. Ref. [18] used a new deep learning-based identification method for fishing gear types from the AIS data, and proposed a convolutional neural network model for the classification of gear type from real-time AIS and environmental data, which would enable the estimation of fish catches from fishing activities, and the detection of IUU fishing and overfishing. Ref. [16] assessed the spatio-temporal impacts of fishing activities in the Northern Adriatic Sea using a fusion of three data sources: terrestrial AIS, corresponding fish catch reports, and environmental variables (sea surface temperature (SST), chlorophyll-a and sea surface Height (SSH)) with a machine learning approach to predict catch per unit effort (CPUE) (a key indicator of fisheries resource exploitation) for effective fisheries management.

Most studies that have used fishing vessel trajectory data have focused on determining the fishing gear type or fishing activity status. It is very difficult, however, to identify the fish species and the quantity of fish caught. Therefore, similar to [8], we used fish landing and trajectory data of coastal and offshore fishing vessels, and proposed a method to discriminate vessel activity into port entry and exit, fishing, and navigation.

## 2. Materials and Methods

### 2.1. Characteristics of the Fishery in the Study Area

In the waters surrounding the island, various water masses, such as Tsushima Middle Water, Tsushima Surface Water, Yellow Sea Bottom Cold Water, and the China Coastal Water, merge, depending on the season. Due to the characteristics of these water masses, many fish species have altered when they appear to access the more suitable fishing grounds [19]. These waters have several resident fish species, and excellent habitats and conditions for spawning and fishing grounds. Therefore, various fishing activities, including fish farming, are carried out in this area for the exploitation of fisheries and other biological resources.

Daily landing reports were obtained from the Korean Fishery Union, which receives and manages fish landing data from fishing ports located on Jeju Island. The dataset is comprised of daily landings for 181 fish species harvested by 771 fishing vessels made up of 125 purse seine, 108 longline, 313 gill net, 164 squid jigging, and 61 trawl vessels from January to December 2018. In Korea, 33 fishing operations exist under the coastal and offshore fisheries, including sectional fisheries. However, the major fishing operations are trawl, purse seine, gillnet, angling, stow net, and trap [3]. In the present study, we focused

on daily fish landing data from purse seine, longline, gill net, squid jigging, and trawl (the major fishing gears extracted from the dataset).

Major fish species that are targeted by these gears include mackerel (*Scomberomorus koreanus*), largehead hairtail (*Trichiurus lepturus*), small yellow croaker (*Larimichthys polyactis*), anchovy (*Engraulis japonicus*), red tilefish (*Brachiostegus japonicus*), red-banded lobsters (*Metanephrops thomsoni*), and brown croakers (*Miichthys miiuy*). Multiple target species were harvested by each gear, as shown in Table S1 (over 95% of the mackerels were harvested by the purse seine, while the longlines harvested the largehead hairtails). The table shows the weights of the total monthly fish landings of target species, indicating that gill nets target small croakera, while trawlers target red-banded lobsters.

### 2.2. Fishing Vessel Trajectory and Fish Landing Data Aggregation

Fishing vessels transmit their trajectory data using AIS and VMS systems. AIS is an automatic tracking device used by vessels to send their navigation status to all vessels nearby at sea. AIS sends unencrypted radio signals that are broadcast publicly as a security feature so that vessels can avoid collisions, and helps authorities to monitor vessel traffic. It is required for fishing vessels with a total tonnage greater than 300, and each broadcasted AIS message contains the ship's ID, equipment type, speed, course, and location, and the signal can be transmitted every few seconds [20]. For safe navigation, countries in the European Union (EU Dir 2011/15/EU) recently began requiring fishing ships of more than 10 m in length to have an AIS transponder [12].

Fishing fleet VMS is owned and operated by individual governments to monitor national fishing fleets and foreign vessels fishing within their waters. The signals transmitted by vessels are encrypted using MF/HF, VHF-DSC, or satellite telecommunications. This means that only the government and those they share it with can access the data. Governments using VMS generally require it for vessels with less than 300 gross tonnage, which means that most of the fishing fleet is visible to VMS. However, neither AIS nor VMS fishing vessel trajectory data include information on the spatial and temporal distribution of fish catch records and vessel owners, nor weigh fish catch at hauling, nor do they record the time and place of operation. Therefore, it is difficult to how much is caught in which location.

In Korea, fishermen report total fish catch and species composition data at the ports to the fishery unions, which collate and forward the same to the government. This includes species, fish state (dry, refrigerated, frozen, alive), port of sale, total weight of catch, and price. Table S2 presents a portion of the fish landing data as described above.

Using the proposed method, we collected AIS-based fishing vessel trajectory data and fish landing data from the Korean Fishery Union over one year (January to December 2018) in the southern part of the Korean waters around Jeju Island. The AIS static data included the 'fishing ship' category of code '30'; however, the fishing vessel type information was not present. Therefore, we extracted the fishing vessel type information of the corresponding fishing vessel based on the MMSI (maritime mobile service identity) ID of the AIS from the Korean Fishing Ship Register Database, which is managed by the National Federation of Fisheries Cooperatives of Korea. In the experiments, we grouped fishing gear types for five fishing operations (purse seine (17%), longline (22%), gill net (29%), squid jigging (14%), and trawl (18%)), which are the most abundant fishing vessels that operate in the waters of Jeju island. Figure 1 represents the experimental procedure for this research.

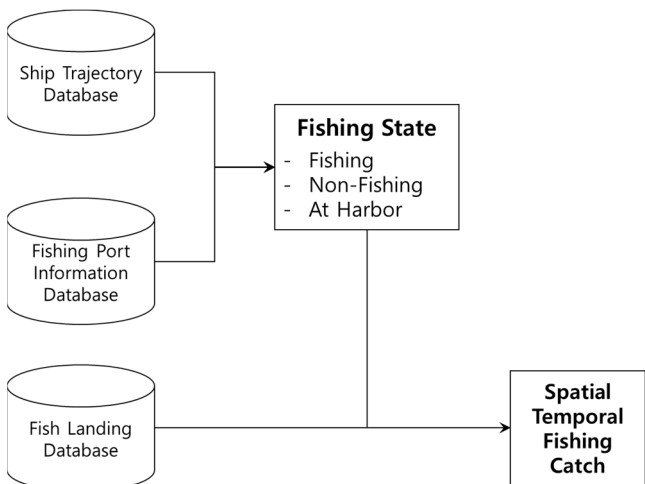

**Figure 1.** The experimental procedure for the proposed method.

First, we extracted the AIS data for vessels in the type '30' category, and sorted by the fishing vessel MMSI ID. Next, each fishing vessel type was checked from the Korean Fishing Ship Register Database for the fishing gear type based on each MMSI ID. We classified the fishing vessel's state into fishing, non-fishing, and port arrival. The fishing activity discrimination was based on average speed profiles for 1 h with a predefined fishing vessel speed range of inactivity. To determine port arrival, we set the location range of the port in advance, and checked fishing vessel's GPS location is within the port area for more than 12 h. Vessels enter and leave the port at any time due to supply, crew change, bad weather conditions, etc. For our work, we identified vessels that frequently trade at the ports and collected data on landing area, method, and time at the fishing harbor using monitoring surveys. Most fishing vessel sales were made in direct connection with the port facilities, and it was confirmed that this takes an average of 10 h or more. Based on the survey results, and as established in this study, the fishing vessel's location was determined to operate at the port for more than 10 h within the designated port area range. When the set conditions were satisfied, the average value of the quantity sold, the catch matching the fishing trajectories, entry, and departure time for the selected gear types were applied as described in the next section. Table S3 shows the fishing status analysis results in a squid jigging fishing vessel trip that operates mostly at night by using the proposed method.

*2.3. Fishing and Non-Fishing Activity Identification*

Classification of fishing activity based on trajectory data is essential for identifying fishing grounds. Here, fishing ground refers to the area where fishing vessels deploy fishing gear to catch fish, and the quantity of fish caught becomes the total catch per vessel per trip. Since catch data are rarely recorded by vessel owners by location and time, we identified the vessel status using trajectory data and fishing operation characteristics. Fishing vessels have several fishing trajectory patterns, depending on the target species and gear. Given that it is difficult to collect navigation and operational characteristics data for all fishing operations, we considered navigation patterns for purse seine, longline, gill net, squid jigging, and trawl fishing vessels for this study.

Usually, fishing vessels travel at low speeds when they are engaged in fishing activities. During non-fishing activities, they sail at high speeds or are stationary when they are in the harbor. Ref. [21] suggested identifying three types of fishing activities: setting, hauling, and non-fishing activities (streaming and wait time) of longlines, and used the vessel speed range characteristics for each activity: setting (4–6 knots), hauling (2–4 knots), and others (less than 2 knots or greater than 6 knots). Ref. [22] observed trawl fishing vessels with speeds of 2 to 4 knots during fishing activities, and more than 6 knots during non-fishing activities We adopt similar methods used by these authors in determining vessel status by using speed profile information obtained from the Fishing Gear of Korea guide report,

issued by the National Fisheries Research and Development Institute. Table S4 presents the characteristics of the speed profiles for the target fishing vessel types.

*2.4. Spatio-Temporal Distribution of Fish Catches*

Combining fishing vessel trajectory data and fish catch data to understand the interrelations between vessel activities and impact is plausible using developing technologies and methods that establish the interconnection between these datasets and provide information that informs management of fisheries resources and marine planning [17]. We proposed a method that allocates catch data to fishing grounds by extracting fishing trajectories and redistributing the catch uniformly to these trajectories, and creating a fine scale spatio-temporal map of catch distribution in the study area. Figure 2 represents the process as detailed above.

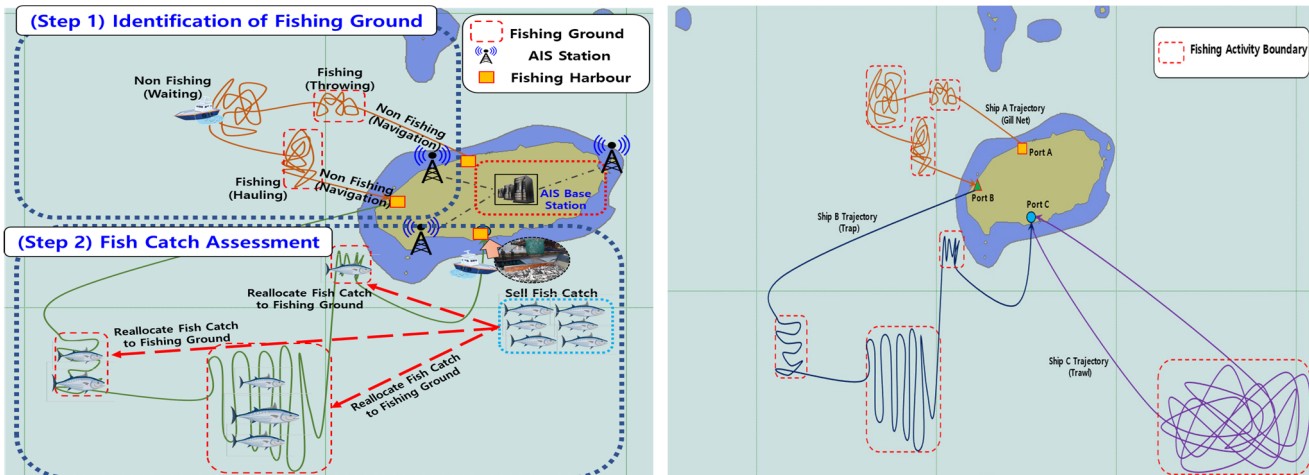

**Figure 2.** The process of extracting fish catches from ship trajectory and reallocating to fishing ground.

To reallocate catch data to fishing trajectories, we established fishing vessel status, i.e., fishing, non-fishing, port entry, and exit. Here, fishing refers to the throwing, hauling, and towing (deployment and retrieval) of fishing gear (Figure 3). Non-fishing refers to the movement between the fishing harbor and fishing ground, and waiting periods after deploying fishing gear. Port entry refers to vessel entering a port to land and sell their catch and port exit refers to vessels steaming out of port to fishing grounds.

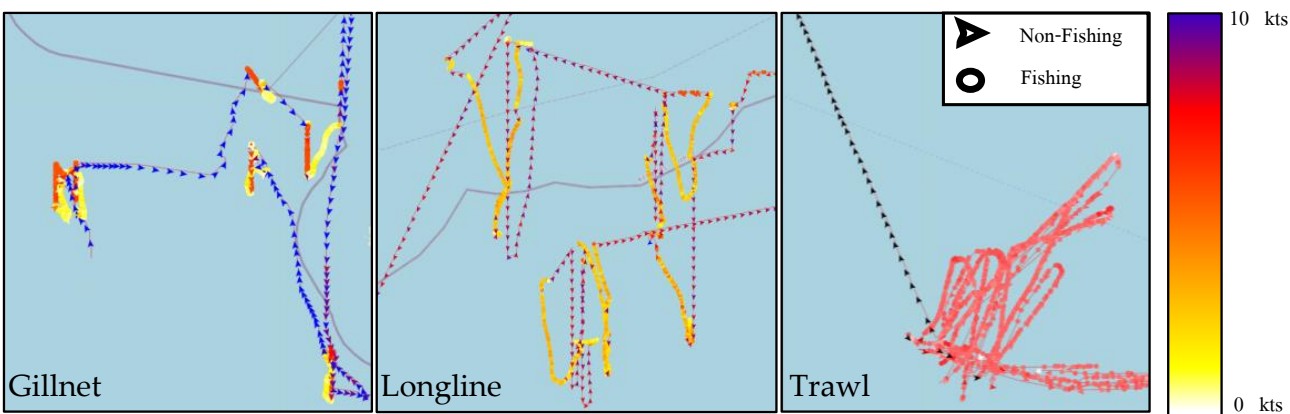

**Figure 3.** Some examples of fishing trip trajectories and vessel activity ("Fishing" or "Non-fishing") for different gear types.

### 2.5. Estimation of Spatio-Temporal Distribution of Fish Catch Data

To estimate the spatio-temporal distribution of catch, we identified the locations of fishing activity, the number of fishing hauls (*NFH*), and the corresponding fish catch information per fishing trip. The total catch is divided by the number of fishing hauls, each of which is allocated to the fishing segment of the trajectory. For a fishing vessel which has a *k* fishing gear type selling fish species *fs* of *m* kg, we propose the equation of catch, *FC* in each fishing *area*, in a given trajectory, at a given time, *t* and for a given species, which is as follows:

$$FC[area, \, t, \, k, \, fs] = \frac{m[k, fs]}{NFH}$$

To explain with specific examples, a fishing vessel entered the port on 8 January 2018, and sold out 120 tons of largehead hairtail. After vessel status determination, if this fishing vessel is found to be operating three locations, the unit fish catch of approximately 40 tons of hairtail shall be allocated to each fishing location. Table 1 shows catch distribution using this method.

**Table 1.** Result of catch redistribution to fishing area.

| Date Time | MMSI [†] | Fishing Ship Type | Latitude | Longitude | Fish Species | Amount (kg) |
|---|---|---|---|---|---|---|
| 13 February 2013 21:14 | 440 | squid jigging fishing | 33.5 | 126.2 | Anglerfish | 7.9 |
| 13 February 2013 21:14 | 440 | squid jigging fishing | 33.5 | 126.2 | Halibut | 12.1 |
| 13 February 2013 21:14 | 440 | squid jigging fishing | 33.5 | 126.2 | Others | 61.5 |
| 13 February 2013 21:14 | 440 | squid jigging fishing | 33.5 | 126.2 | Halibut | 38.9 |
| 13 February 2013 21:14 | 440 | Gillnet | 33.5 | 126.2 | Perch | 21.3 |
| 13 February 2013 21:14 | 440 | Gillnet | 33.5 | 126.2 | Croaker | 76.9 |
| 13 Februray 2013 21:14 | 440 | Gillnet | 33.5 | 126.2 | Jacopever | 1.2 |
| 13 February 2013 21:14 | 440 | Gillnet | 33.5 | 126.2 | Ray | 0.6 |
| 13 February 2013 21:14 | 440 | Purse Seine | 33.5 | 126.2 | Mackerel | 115.2 |
| 13 February 2013 21:14 | 440 | Gillnet | 33.5 | 126.2 | Croaker | 342.6 |
| 13 February 2013 21:14 | 440 | Purse Seine | 33.5 | 126.2 | Sea eel | 31.8 |
| 13 February 2013 21:14 | 440 | Gillnet | 33.5 | 126.2 | Tile fish | 75.3 |

[†] The maritime mobile service identity (MMSI) is a unique identifier assigned to a vessel. Complete numbers are not shown to maintain vessel anonymity.

## 3. Results

### 3.1. Mapping Spatio-Temporal Distribution of Catch

Fishing vessel trajectories and a set of standards were applied to identify individual fishing trips, as described in Section 2. The standards were adapted for the five types of fishing gear considered during this study to enable the assessment of spatio-temporal variations in their activities. To assign catch values to fishing trajectories, the area under study was divided into square grid cells. The study area was divided into square grid cells, and the spatial resolution was set at 0.1° (5° by 5° pixel for each 0.5° grid cell) and 0.2° (2.5° by 2.5° pixel for each 0.5° grid cell) for fishing vessel trajectory and catch distribution (Figures 4–7). The calculated fish catch values were assigned to each grid cell to generate maps of reallocated fish catch distribution for each gear type (Table 1). The resulting maps show distributions of catch values to fishing trajectories and highlights spatio-temporal variations in fish catch amounts and intensities of fishing vessel activity. We produced vessel trajectory and catch distributions mapped at fine spatial scales and show variations in fishing activities and reliance on fishing grounds and variations in locations fished by fishing gears. Figure 4a–e shows the fishing vessel trajectory distribution, while Figure 5a–e shows the fish catch distribution as computed by the proposed method. The

trajectory maps show that the areas fished varied for each gear type. Vessels operating purse seine and squid jigging gears were dependent on nearshore areas, and occasionally moving to areas further offshore (Figure 4a,d). Fishing vessels that operate longlines, gill nets, and trawl nets operated within fishing grounds in locations further offshore, with longline vessels and gill net vessels operating from near coastal areas and moving offshore to the northern, western, and southern areas of the island. Trawlers, on the other hand, showed restricted operations in the south and northeastern parts of the island.

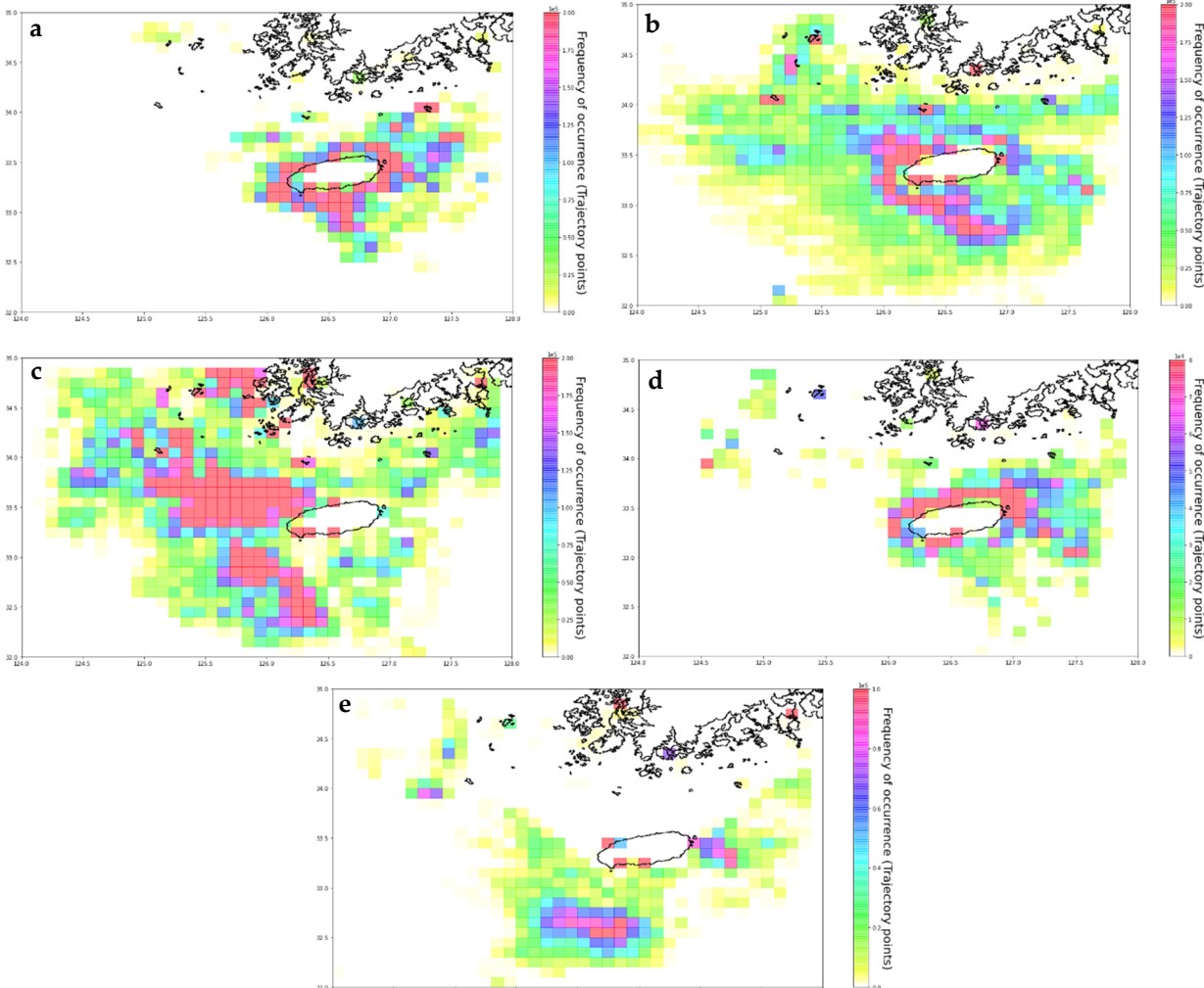

**Figure 4.** Map of fishing vessel trajectory distribution in the study area for (**a**) purse seine, (**b**) longline, (**c**) gill net, (**d**) squid jigging, and (**e**) trawl fishing gears throughout January to December 2018.

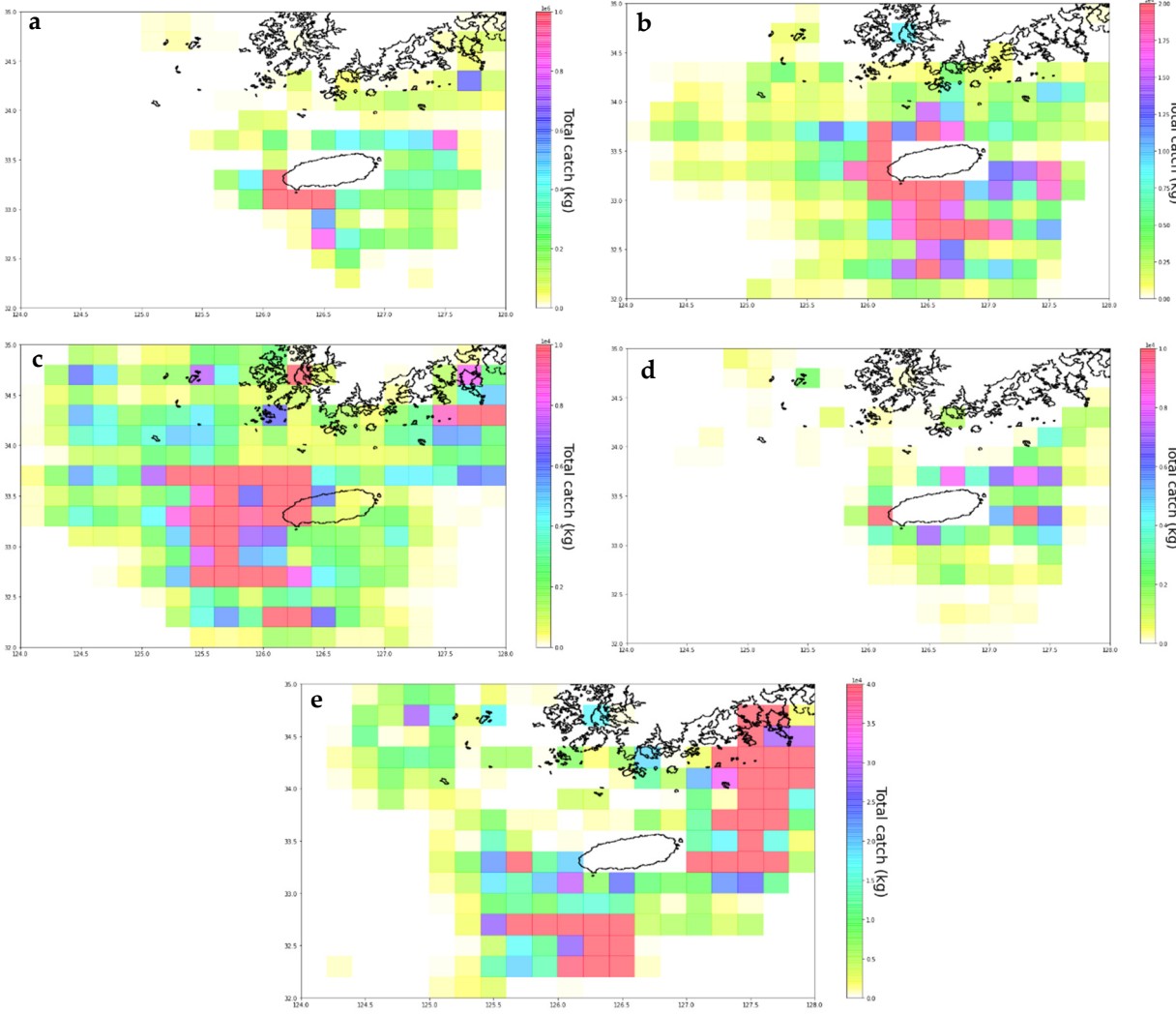

**Figure 5.** Map of catch distribution in the study area for (**a**) purse seine, (**b**) longline, (**c**) gill net, (**d**) squid jigging, and (**e**) trawl fishing gears throughout January to December 2018.

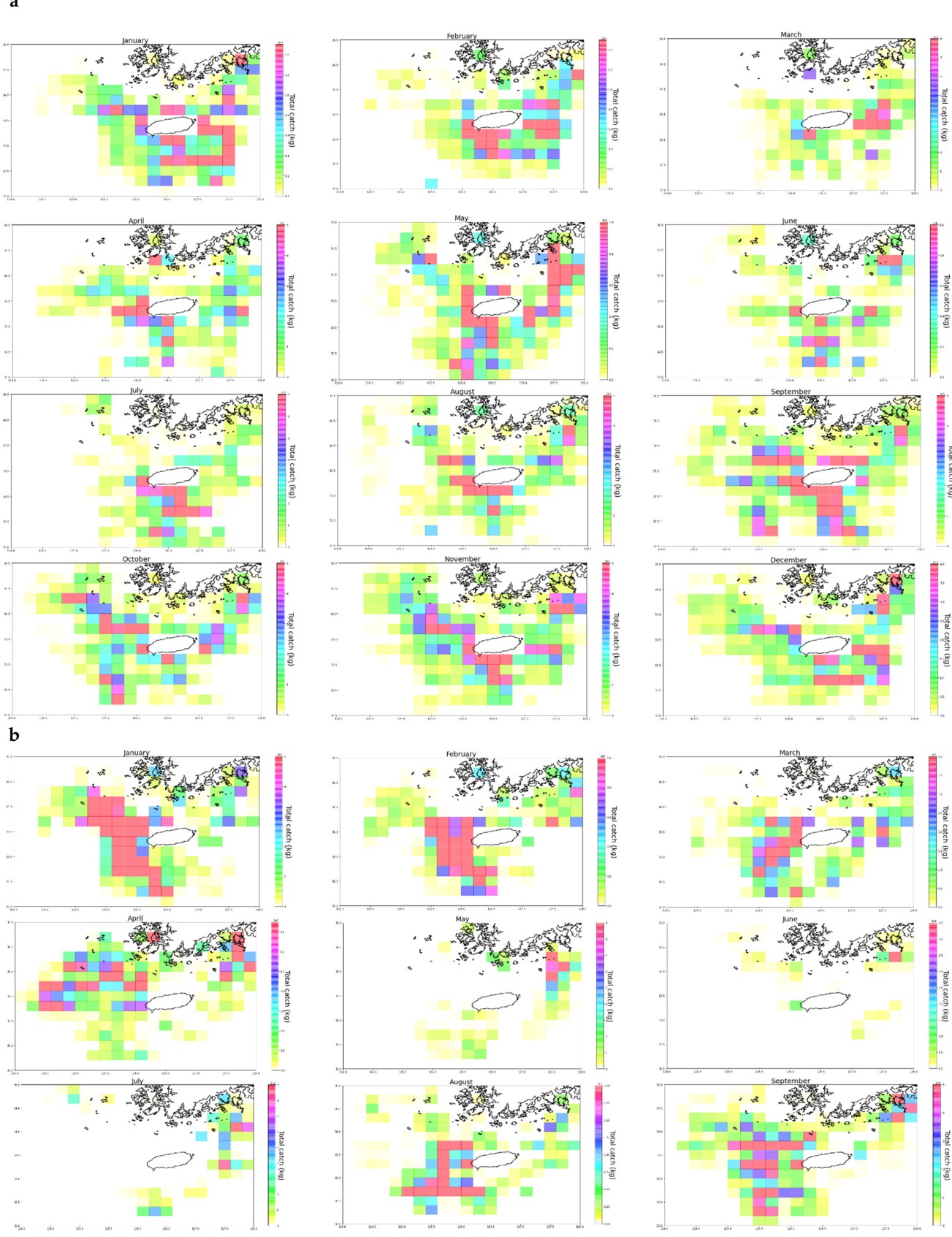

**Figure 6.** *Cont.*

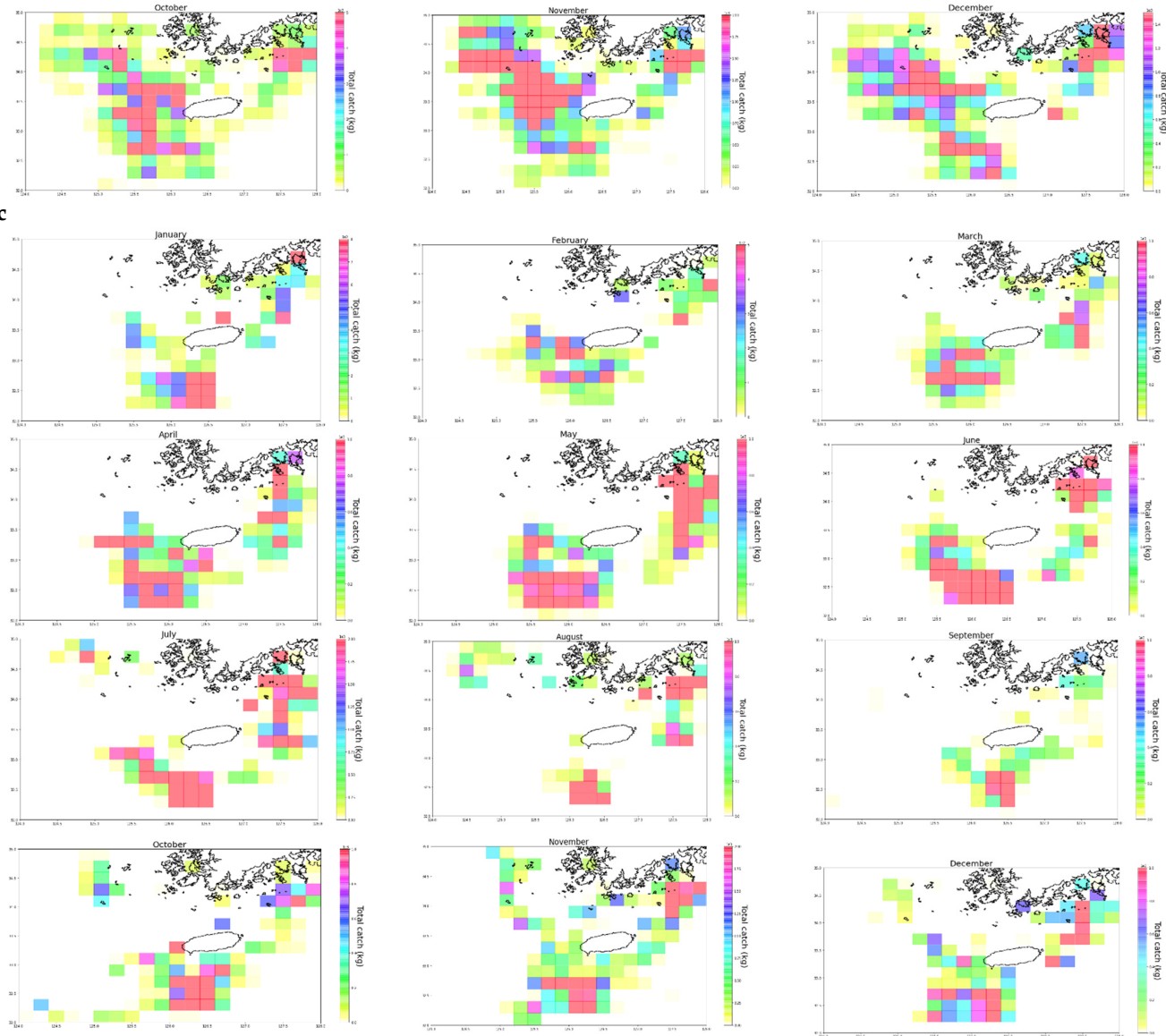

**Figure 6.** Map of spatio-temporal distribution of (**a**) largehead hairtail, (**b**) small yellow croaker, and (**c**) red-banded lobster in the study area throughout January to December 2018.

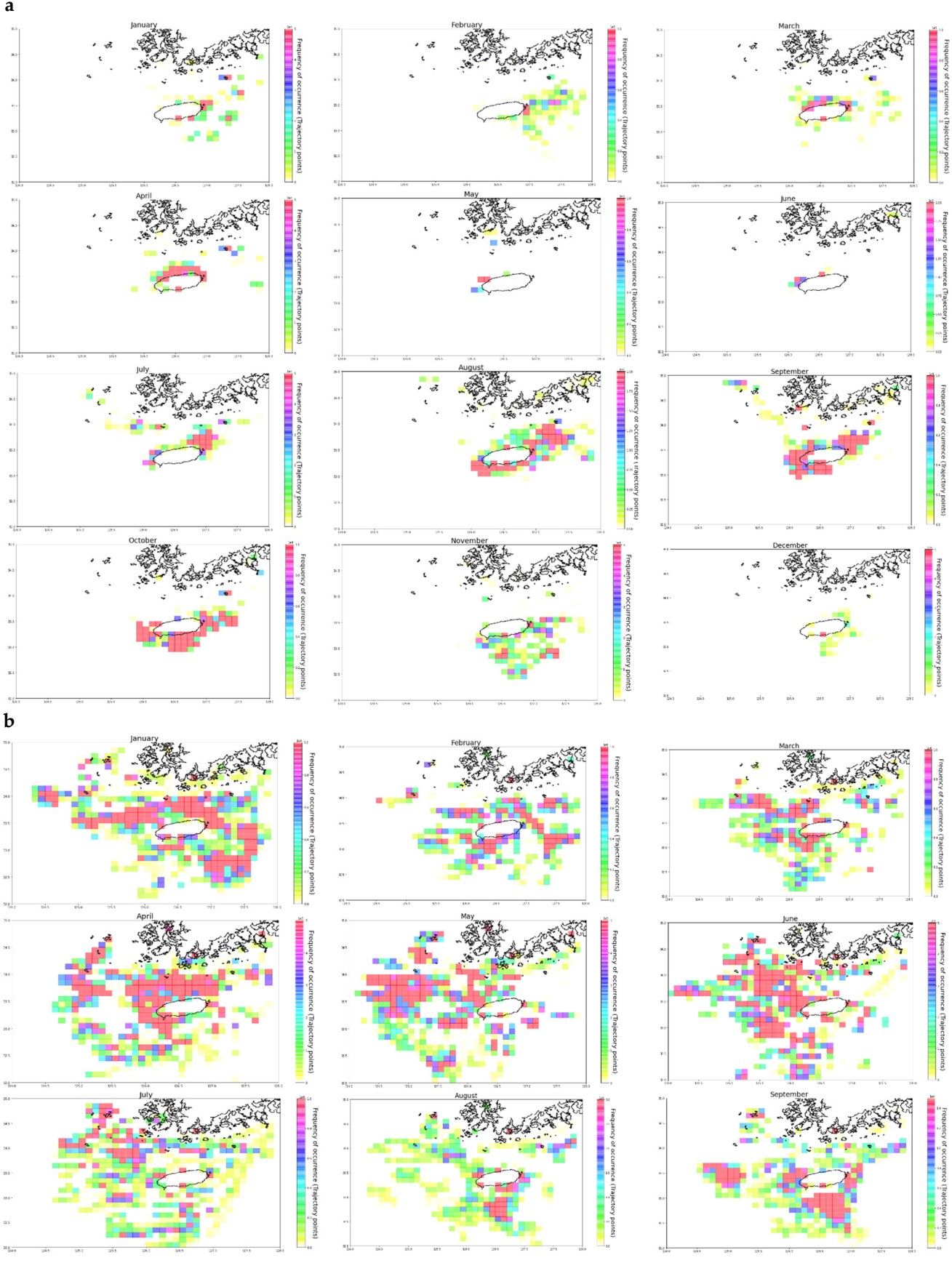

**Figure 7.** *Cont.*

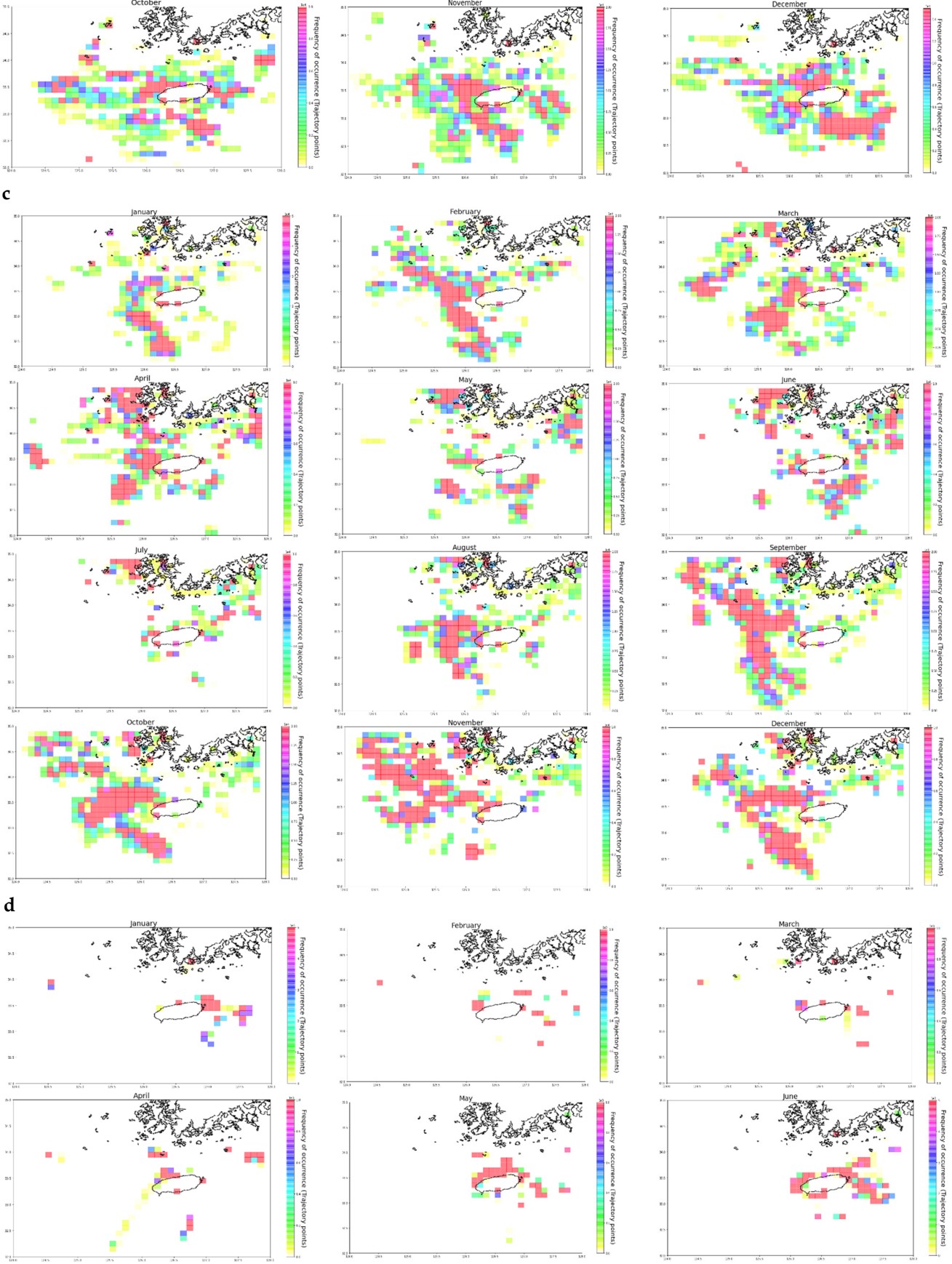

**Figure 7.** *Cont.*

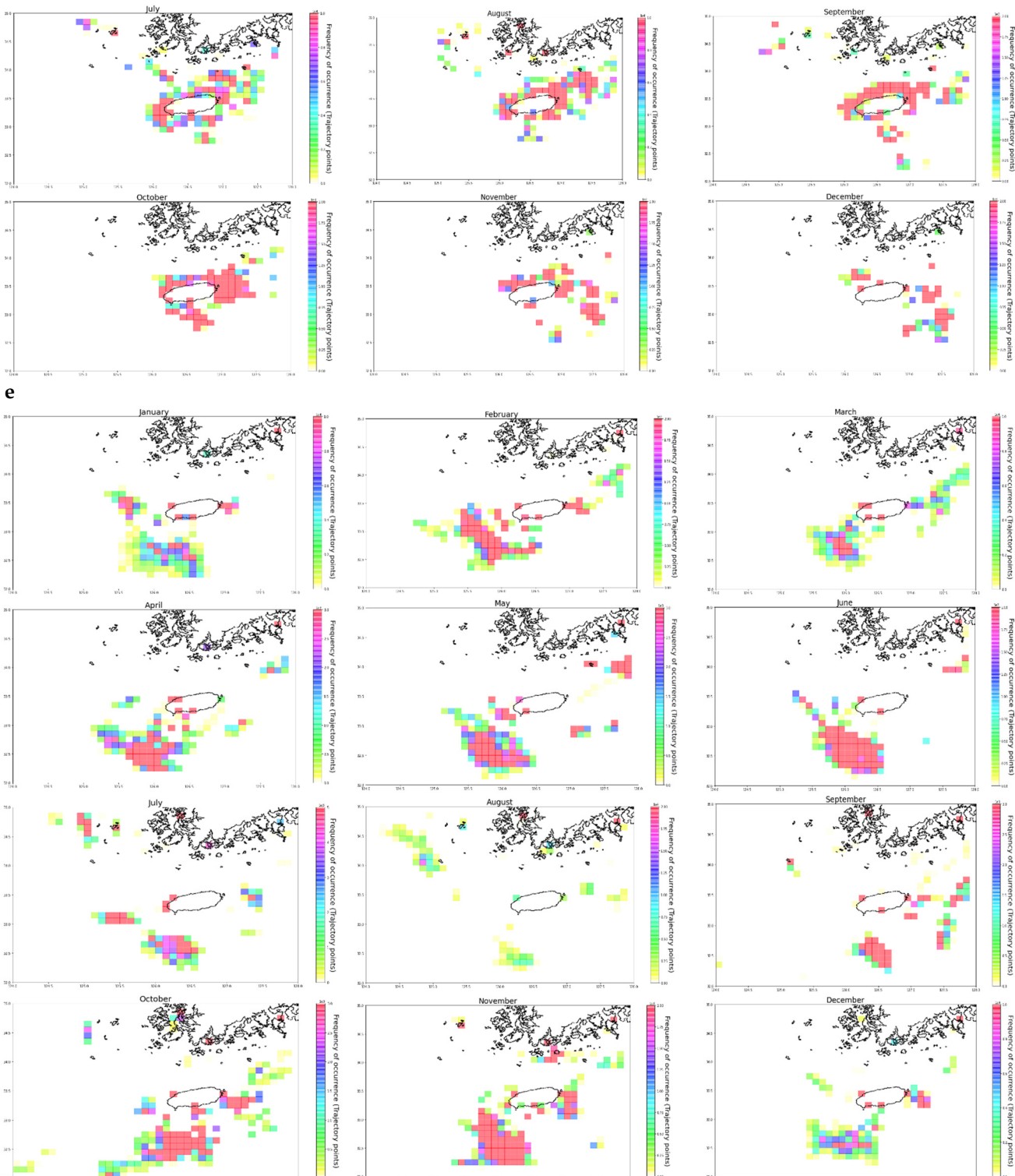

**Figure 7.** Spatial and temporal distribution of fishing gear activity of (**a**) purse seine, (**b**) longline, (**c**) gill net, (**d**) squid jigging, and (**e**) trawl fishing gear types in the study area throughout the period January–December 2018.

Differences in activities of vessels and gears deployed are seen in the distribution map provided by fishing vessel trajectory data. Vessels operating longlines, gill nets, and trawls showed a relatively greater extent of activity. Fishing trips involving these vessels are known to be longer, in terms of time spent and distance traveled [3]. The degree of reliance on fishing grounds varied among gear types and locations where they were operated.

Generally, the number of vessels and the degree of reliance on fishing locations were directly related, i.e., locations with higher number of vessels showed greater dependence. However, despite the number of longline vessels being three times greater than gill net vessels, a similar degree of distribution was observed for the two vessel types. In addition, longline vessels had a greater dependence compared to purse seine and squid jigging vessels, despite having similar number of vessels.

Using the proposed method, we can show specific areas where fishing activity occurred and indicate the amount of catch obtained from these locations (Figure 5). The value for each cell is the sum of all catch amounts calculated for fishing trajectories that occurred throughout January–December 2018 within the cell. The resulting maps show catches generated separately for each gear type. As discussed in Section 2.1, each gear differs in target species and the quantity of catch harvested. Each gear type is associated with a single most abundant species, as shown in Table S1. The spatio-temporal distribution of the fishing grounds for the largehead hairtail, small yellow croaker, and red-banded lobster are shown in Figure 6. These species are targets for longlines, gill nets, and trawl nets, respectively (Table S1). Therefore, the catch distribution maps generated give a detailed view of the spatio-temporal distribution of these species, while indicating catch amounts taken from specific locations.

The spatial pattern of fishing intensity varied between January and December for hairtail, small yellow croaker, and red-banded lobster. The hairtails catch was distributed throughout the entire area, but its abundance was higher during the summer through to the winter. The small yellow croaker was observed to be predominantly distributed in the northwest and southeast throughout the year except for May to July, when catches were absent. Most of the fisheries are known to overlap in terms of fishing grounds and target species, regardless of the fishing gear and method. The gill net, which targets small yellow croakers, also harvested other fish species, as it is observed to be active in the northeast and southeast between the months of May to July, as illustrated in Section 3.2, Figure 7. Catches of hairtail and red-banded lobster were observed throughout the year, as is consistent with their corresponding fishing operations (longlines and trawl, respectively).

*3.2. Assessment of Fish Resource and Vessel/Gear Activity Distribution*

Average fishing activity per gear type exhibited monthly variations during the study period (Figure 7). For the purse seine gear, from January to April and June to July, most fishing activities focused on the northern side of the island, while from August to November, activities were quite uniformly distributed over the study area. Low fishing activity was experienced in May and in December (Figure 7a). Longline fishing activities occurred on all sides of the island from November to April, and concentrated near the southern side from June to October (Figure 7b). For gill net fishing, with the exceptions of June and July, activities were restricted to the north, northwest, and southern sides of the island, with activities spreading out northwards during August to December (Figure 7c). Squid jigging fishing activities occurred on all sides of the island, with increase in spread and intensity generally from January to December (Figure 7d). For trawl net, fishing activities were mostly restricted to the southwestern side from January to July, and spread southeastwards from September to December (Figure 7e).

## 4. Discussion

In the present study, the catch variation and species abundance of commercial fishes were investigated in relation to the fishing periods within the Korean Exclusive Economic Zone (EEZ) around Jeju Island [23]. Some studies have identified spatial variations in species abundance and richness, while others have attributed these changes to seasonal environmental effects and oceanographic conditions such as salinity, temperature, and current [24–27]. Ref. [28] investigated the frequency of appearance of fishing vessels on fishing grounds in the waters around Jeju Island using AIS data from October 2016 to October 2017. In this study, fishing vessel distribution indicated a monthly shift in fishing

grounds. High concentrations of fishing vessels in the northwest to the south coast during winter were associated with yellowtail fishery, while those in the northeast during summer were associated with squid jigging and hairtail fishing. The distribution of the yellowtail stock located off the northeast coastal waters was found to be influenced by abiotic factors (temperature and tidal currents) and biotic factors (predator-prey interaction) [29].

We combined fish landing data and AIS data for fishing vessels operating in Korean waters around Jeju island. The classification of fishing activity was performed using a speed profile classification of the Korean Fishing Gear Guide Report. These were similar to the classification described by [21,22]. We assessed the spatio-temporal distribution of fish catch and fishing gear activity using fishing vessel trajectory data, and daily fish landing data using the proposed method described in Section 2. The results indicate that our method identified fishing vessel tracks related to actual fishing activities within fishing grounds, and mapped fishing trajectory patterns and catch distributions at fine spatial scales. We mapped catch distribution for the different gear types, as they differ in their efficiency for harvesting different species [8]. Seasonal variations were observed in the distribution of fish species and patterns of fishing vessel activity, which indicates that the fish species use different marine areas due to various factors, such as environmental factors as indicated by [29]. High-resolution maps generated using the vessel trajectory data improve the understanding of spatial and temporal extent of fishing vessel activity and their effects at the ecosystem level. By combining catch data and AIS data, we understand that this could be useful in setting appropriate management measures at the métier level.

Tracking fishing vessel activities makes it possible to characterize fishing trips with respect to fishing gear types, and to identify long-term patterns that will help understand the changes in species abundance and distribution and predict potential changes in fishing vessel activities. Regulators can draw from the relevance of AIS data in providing a real-time science-based approach that can enhance laws that define limits on fishing vessels, fishing method and gear, resource protection and recovery, and catch limits, and reinforce appropriate input/output controls and technical measures that regulate the fisheries. The potential of AIS data in providing solutions to conflicts arising from the competing exploitation of fish resources by coastal and offshore fishing vessels, given that most of the fishing grounds overlap [3], stems from the open-access nature of the data generated and the availability of technology and tools that can provide real-time solutions and with predictive power to inform management measures that can safeguard fish stocks and increase the resilience of the fisheries industry.

The importance of the fusion of AIS and fish landing datasets has been emphasized given the opportunities provided by trajectory data for fisheries research and the insights they can provide to guide fishery management decisions [30]. However, the inclusion and utilization of environmental data provide a more holistic and realistic assessment of the impacts, and aid in predicting future impacts and events [8]. The biological and physico-chemical components of the marine environment and their interactions affect fish abundance and distribution, and influence the patterns of observed fishing activities. The nature of the different habitats and physical ocean environment coupled with oceanic processes, such as currents and fronts, create the balance that supports the survival and distribution of fish species, food availability, and trophic interactions. Thus, the observed patterns of fishery distribution and the association with seasonal variations emphasize the need to combine environmental variables such as sea surface temperature (SST), chlorophyll-a, sea surface height (SSH), and currents in future analysis. This method demonstrates that a fusion of these datasets provides a good opportunity to predict future impacts, improve management and sustainable exploitation of resources, and inform marine spatial planning within the context of climate change [8,30].

This study presents the initial results for the combined use of vessel trajectory data and fish landing data to assess the spatial and temporal patterns of distribution for gear and fishing vessel activity, and their level of dependence on fishing grounds. The method proposed in this study was useful in identifying reference points for major gears and in

improving our ability to investigate seasonal variations and make predictions of potential changes in fishing activities of vessels in the coastal and offshore fishing industry. We used a short temporal extent of one year for both AIS and fish landing dataset in our research to analyze the spatial and temporal extent of fishing activities. The current work did not incorporate some variables such as environmental datasets such as SST, salinity, and wind, which can help assess multifactor impact on the marine environment. With access to datasets collected over a longer period and by applying available machine learning methods [8,16], we can analyze impacts of climate change on fishing activities, provide a more extensive assessment of seasonal changes in catch and effort, and make predictions that can support sustainable resource management and marine spatial planning [30].

**Supplementary Materials:** The following are available online at https://www.mdpi.com/article/10.3390/su132413841/s1, Table S1: The weights of total fish landing at Jeju fishing harbors from January to December 2018, Table S2: Fish landing data for the waters surrounding Jeju island, Table S3: Example showing analysis results of allocating fishing status to a vessel during a fishing trip, Table S4: Speed profiles for different fishing activities.

**Author Contributions:** Conceptualization, S.A.O. and K.-I.K.; methodology, K.-I.K.; software, K.-I.K.; validation, S.A.O. and K.-I.K.; formal analysis, S.A.O. and K.-I.K.; investigation, K.-I.K.; resources, K.-I.K.; data curation, S.A.O. and K.-I.K.; writing—original draft preparation, S.A.O. and K.-I.K.; writing—review and editing, S.A.O. and K.-I.K.; visualization, S.A.O. and K.-I.K.; supervision, K.-I.K.; project administration, K.-I.K.; funding acquisition, K.-I.K. All authors have read and agreed to the published version of the manuscript.

**Funding:** This research was funded by the Basic Science Research Program of the Research Institute for Basic Sciences (RIBS) of Jeju National University through the National Research Foundation of Korea (NRF) funded by the Ministry of Education (Grant no.: 2019R1A6A1A10072987) and was a part of the project titled, "Development of AI Based Smart Fisheries Management System" which is funded by the Ministry of Oceans and Fisheries, Korea.

**Conflicts of Interest:** The authors report no conflicts of interest.

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
