# Peer review of "Spatio-Temporal Fish Catch Assessments Using Fishing Vessel Trajectories and Coastal Fish Landing Data from around Jeju Island"

_sustainability, doi:10.3390/su132413841_

Round 1
Reviewer 1 Report
General comments
The current version of the manuscript is significantly improved compared to the first submission. Please see the amendements and suggestions reported in my specific comments below.
Specific comments
Page 1, lines 8-11. Consider rephrasing as follows: “While pressure on marine ecosystems leading to declines in global fish catches have been attributed to excessive fishing and to unregulated and unreported fishing, existing management practices have yet to fully address these declines”.
Page 5, line 219 and page 6, line 241. Switch the order of appearance of Table S3 and S4 in the text, and change their occurrence in the supplementary material as well. So, Table S3 should be “Example showing the analysis results ALLOCATING fishing status TO A VESSEL DURING a fishing trip” (please note the suggested amendments in the caption) and Table S4 “Speed profiles for different fishing activities”.
Page 6, line 244. In the current version of the manuscript Figure 2 (“Different gear types showing fishing trip trajectories and vessel activity”) is cited in the text after Fig. 3. I suggest to switch the order of appearance of the two figures, changing their names accordingly (Fig.2->Fig.3 and Fig.3->Fi.2). I also suggest a new caption for this figure: “Some examples of fishing trip trajectories and vessel activity (“Fishing” or “Non-fishing”) for different gear types.”
Page 6, line 253. Replace in the text the wording “Figure 3” with “Figure 2” (“The process of extracting fish catches from ship trajectory and reallocating to fishing ground.”).
Page 6, line 257. Replace in the caption text the wording “Figure 3” with “Figure 2” (“The process of extracting fish catches from ship trajectory and reallocating to fishing ground.”). This figure should precede the one about the examples of fishing trip trajectories, which should become Fig. 3 instead.
Page 6, line 261. Replace in the text “Fig. 2” with “Fig. 3”, that is, after the amendment (switch of order) the one showing the examples of fishing trip trajectories.
Page 8, lines 293-295. Though the quality of Figs 4-5 is quite low (dpi in the image files should be significantly increased) , the apparent geographical resolution of Fig. 4 (vessel trajectory) and Fig. 5 (catch distribution) is 0.05° (10x10 pixel for each 0.5° grid cell) and 0.1° (5x5 pixel for each 0.5° grid cell), respectively. Please modify the text accordingly, introducing such an information.
Page 8, line 301. Replace “show” with “shows”.
Page 8, line 302. Replace “show” with “shows”.
Page 8, lines 312-315. Insert minimum and maximum values in the color bar (I guess the variable should be “counts”, i.e. frequency of occurrence of a vessel in a certain lat/lon position). Complete the caption as follows: “Map of fishing vessel trajectory distribution in the study area for a) purse seine, b) longline, c) gill net, d) squid jigging and e) trawl fishing gears throughout January to December 2018”.
Page 8, line 318. Consider replacing “operations” with “trips”.
Page 9, lines 328-331. Insert minimum and maximum values (expressed in units of catch weight, I guess) in the color bar. In addition, please note that the palette of colors shown in the colorbar is different from colors of “pixels” in Fig. 5a-e. Complete the caption as follows: “Map of catch distribution in the study are for a) purse seine, b) longline, c) gill net, d) squid jigging and e) trawl fishing gears throughout January to December 2018”.
Page 9, lines 334-335. Complete description as follows: “…trajectories that occurred throughout the period January-December 2018 within the cell.”
Page 9, lines 344-345. Insert color bar, with minimum and maximum values (expressed in units of catch weight, I guess; please specify). In the caption, specify the time period considered (January-December 2018).
Page 9, lines 349-353. Consider amending as follows: “The hairtails catch was distributed throughout the entire area, but its abundance was higher during the summer through to the winter. The small yellow croaker was observed to be predominantly distributed in the northwest and southeast throughout the year except from May to July, when catches were absent.”.
Page 10, line 359. The authors should clarify what they mean with “fishing gear activity” in this section (and in Fig. 7 as well). Is actually what is mapped in Fig. 7 the spatio-temporal distribution of total estimated fish catch by gear type (and by month)? If yes, consider rephrasing the title of this section as follows: "Assessment of temporal distribution of total fish catch by main fishing gear type". As an alternative, the title of this section could be removed, considering that, if “fishing gear activity” shown in Fig. 7 is actually “fish catch”, this section would be again about the mapping of spatio-temporal distribution of catch (which is exactly the title of section 3.1, by the way).
Page 10, line 361. Improve the quality (dpi) of the image, possibly also giving more evidence to the coastline.
Page 10, lines 361-364. Consider rephrasing as follows: “For the purse seine gear, from January to April and from June to July most fishing activity focused on the northern side of the island, while from August to November activities were quite uniformly distributed over the study area. Low fishing activity was experienced in May and in December (Fig. 7a).
Page 10, lines 364-365. Complete as follows: “…near the southern side from June to October (Fig. 7b).”.
Page 10, lines 367-368. Complete as follows: “…with activities spreading out northwards during August to December (Fig. 7c).”.
Page 10, line 369. Complete as follows: “…and intensity generally from January to December (Fig. 7d).”.
Page 10, line 371. Complete as follows: “…spreading southeastwards from September to December (Fig. 7e).”.
Page 10, lines 376-377. Complete caption of Fig. 7 with “...throughout the period January-December 2018.”. What do you mean with “fishing gear activity” here? Please clarify which variable is mapped, is it catch (in weight) or fishing vessel positions (frequency of occurrence)? Insert a color bar accordingly.
Page 11, line 403. Consider replacing “areas of waters” with “marine areas”.
Page 11, line 416. Consider replacing “competing use” with “competing exploitation of fish resources”.
Page 11, line 420. Consider replacing “resource” with “fish stocks”.
Page 12, line 444. Consider replacing “data” with “dataset”.
Page 12, line 450. Consider replacing “inform” with “support”.
Reviewer 2 Report
See attached file.

Author Response
Please see the attached.

This manuscript is a resubmission of an earlier submission. The following is a list of the peer review reports and author responses from that submission.
Round 1
Reviewer 1 Report
The manuscript “Spatio-temporal fish catch assessments using fishing vessel trajectories and coastal fish landing data from around Jeju Island” proposed an interesting method to estimate the spatio-temporal distribution in terms of fish catch, fishing vessel and gear activity using fishing vessel trajectories and coastal fish landing data around Jeju Island. Speed profiles were used to identify fishing and non-fishing activities of vessels targeted different fish species and total fish catch was reallocated to fishing grounds by combining of AIS and fish landing data for January to December 2018. The results suggest that seasonal variations were observed in the distribution of fish species and fishing vessel activity patterns.
Although the study is interesting and provides a methodology for spatio-temporal fishing catch assessments, the manuscript has some challenges.
- First of all, this study averaged the fish landing data into fishing activities. However, the spatial distribution of fishery resources is not uniform. Can the effect of CPUE on fishing catch be considered in this study?
- In this study, theestimated fish catch distribution was based on the fishing vessel trajectory distribution. Therefore, please consider whether is reasonable to verify the proposed results by comparing the fishing vessel trajectory distribution with estimated fish catch distribution.
- Please explain what do the numbers in the grids of figures 3 and 4 represent? In addition, colors in pictures should also be represented by color bars.
- Please check the text “Fig. 8-10”on line 336,which was not found in the article.
- There are no results in this paper to support the impact of marine environment on the abundance and distribution of fish species. Therefore, the impact of marine environment should not be included in the conclusion.
Reviewer 2 Report
This paper describes the fishing ground estimation by AIS data.
Estimates should be made of the genuinely unavailable information. It is clear that the captain and skipper know the exact location of the fishing grounds, and even if VMS and AIS can be used to estimate this information, it is considerably less accurate than the information known by the captain and skipper. I did not understand why it should be estimated by using AIS.
It is obvious that the position of a fishing vessel has a high correlation to the operating position. Only the fishing vessels conduct fishing, and the fishing vessels are the vessels aimed to go to the fishing ground. The results in 6.1. does not prove the validity of this method.
In the introduction, it is mentioned without citation that overfishing and IUU fishing are undermining the recovery of the resource despite the fisheries management efforts being implemented by the Korean Government, but citations are required.
The relationship between this study and stock management and stock assessment is unclear. It may simply be an inaccurate identification of the location of the fishing grounds from AIS, which is already known accurately by the captains and fishers. I cannot imagine how this result provides a good basis for the sustainable utilization and effective management of the fisheries resources.
Reviewer 3 Report
The authors outlined an extensive work trying to reliably approach a high heterogeneous issue that of multi-gear fishery. I consider the basic premise, method developed and questions asked by this study as valuable and interesting. Likewise, the main findings of the study, as summarized in the conclusions, have broad usefulness. I congratulate the authors for approaching successfully this topic, a not easy work to handle due to the presence of different levels of data complexity and availability.Author Response
Please see the attachment.

Reviewer 4 Report
General comments
In general, I found the manuscript original and well written, and potentially may represent an important piece of information in support to fisheries management for operators, policymakers and the government agencies. However, I have some concerns about the clarity in reporting the applied methodology and the presentation of results. In particular, I believe the manuscript would greatly benefit from a deeper description of the method used for the calculation of CPUE values and of their the spatio-temporal mapping. Specifically, in Figs 3-4, reporting spatio-temporal distributions of catch and effort (fishing vessel trajectories) data, legends are missing and colors used in the graphs are not labelled, and this impedes a comprehensive evaluation of results. In addition, the equation used for the evaluations of the fish catch in each fishing ground, reported in line 273, is poorly described, as far it concerns its indexing (k = fishing gear type?; s=?, n=?), its temporal and spatial resolution and the denominator (why in the equation FC is divided by "ship number" PLUS "number of fishing cycles"?).
Specific comments are given below:
Page 4, line 173. Replace "more of" with "most of"?
Page 4, line 186. Replace "patio-temporal" with "spatio-temporal".
Page 6, line 257. Replace "section 5.1" with "section 4.3".
Page 6, line 260. Replace "section 5.2" with "section 4.4".
Page 6, line 273. Improve the explanation of the equation. What does “s” stand for? Which is the spatio-temporal resolution of the calculated values? Why fish catch is divided by “Ship Number AND “Number of Fishing Cycles”?
Page 7, lines 284-287. Please report the percentage on the total for the five fishing gears types considered in the study.
Page 7, lines 300-302. Which is the spatial resolution of fish catch data? How was it established and used in figure 3?
Page 8, line 315. Replace “Figure 3-7” with “Figure 3“?. Please insert a legend for labelling the colors used in the graphs. The right side of figure 3.a, representing fish catch distribution, has a different spatial limits compared with the other sub-graphs.
Page 9, line 336. Replace “Figure 8-10” with “Figure 4”. Please insert a legend for labelling the colors used in the graphs.
Round 2
Reviewer 2 Report
Comment 1
The author described the reason for conducting this research as
- fishermen are hesitant in sharing fishing grounds information with other vessel owners
- this research estimates the fishing activities of the fishing vessels with vessel identifier information.
This revision clearly shows that this research tries to show the fishing activities that fishers do not like to disclose.
I believe the fishing ground distribution should be collected directly from fishers with their consent. Disclosure of the fishing ground without fishers' consent have serious ethical problem.
Comment 2
The results in 6.1 does not prove the validity of this method as I pointed out, but no revision was made in the manuscript.
